

# Non-deep physiological dormancy and germination characteristics of *Primula florindae* (Primulaceae), a rare alpine plant in the Hengduan Mountains of southwest China

Yingbo Qin[1], Boyang Geng[1], Li-E Yang[2] and Deli Peng[1,3]

[1] School of Life Science, Yunnan Normal University, Kunming, Yunnan, China
[2] Faculty of Geography, Yunnan Normal University, Kunming, Yunnan, China
[3] Key Laboratory of Yunnan for Biomass Energy and Biotechnology of Environment, Yunnan Normal University, Kunming, Yunnan, China

Corresponding author
Deli Peng, pengdeli@ynnu.edu.cn

## ABSTRACT

Timing of seed germination is directly related to the survival probability of seedlings. For alpine plants, autumn-dispersal seeds should not germinate immediately because the cold temperature is not conducive to the survival of seedlings. Seed dormancy is a characteristic of the seed that prevents it from germinating after dispersal. *Primula florindae* is an alpine perennial forb endemic to eastern Tibet, SW China. We hypothesized that primary dormancy and environmental factors prevent seeds of *P. florindae* to germinate in autumn and allow them to germinate at the first opportunity in spring. We determined how $GA_3$, light, temperature, dry after-ripening (DAR) and cold-wet stratification (CS) treatments affect seed germination by conducting a series of laboratory experiments. Firstly, the effects of gibberellic acid ($GA_3$; 0, 20, and 200 mg L$^{-1}$) on germination of freshly shed seeds at alternating temperatures (15/5 and 25/15 °C) were immediately investigated to characterize seed with a physiological dormancy component. Then, the fresh seeds treated with 0, 3, and 6 months of after-ripening (DAR) and cold-wet stratification (CS) were incubated at seven constant (1, 5, 10, 15, 20, 25, and 30 °C) and two alternating temperatures (5/1, 15/5, and 25/15 °C) at light and dark conditions. Fresh seeds were dormant, which only germinated well (>60%) at 20, 25, and 25/15 °C in light but not at ≤15 °C and to higher percentages in light than in dark. $GA_3$ increased germination percentage of fresh seeds, and DAR or CS treatments increased final germination percentage, germination rate (speed), and widened the temperature range for germination from high to low. Moreover, CS treatments reduced the light requirement for germination. Thus, after dormancy release, seeds germinated over a wide range of constant and alternating temperatures, regardless of light conditions. Our results demonstrated that *P. florindae* seeds have type 2 non-deep physiological dormancy. Timing of germination should be restricted to early spring, ensuring a sufficient length of the growing season for seedling recruitment. These dormancy/germination characteristics prevent seeds from germinating in autumn when temperatures are low but allow them to germinate after snowmelt in spring.

## INTRODUCTION

The timing of germination is a crucial event in the life cycle of a plant, and it significantly influences the growth and survival of seedlings (*Donohue et al., 2010*). This phenomenon is especially true in alpine regions, where seedlings must attain a critical size by the end of the short growing period to be able to survive the long, harsh winter (*Wang et al., 2017*). Thus, alpine plants have evolved a series of mechanisms to prevent the germination of freshly shed seeds in autumn and to allow seeds to germinate at the first opportunity in spring, so that seedlings grow under optimal environmental conditions and maximize the length of the growing season (*Jaganathan, Dalrymple & Liu, 2015*). Several mechanisms control the timing of germination, including seed internal (*e.g.*, dormancy) and external (*e.g.*, temperature, moisture, and light) factors, with seed dormancy being a major mechanism (*Finch-Savage & Leubner-Metzger, 2006*).

Seed dormancy prevents germination in a specified period, under any combination of environmental factors that otherwise favor germination for non-dormant seeds (*Baskin & Baskin, 2004*). Seed dormancy is a very common phenomenon in arctic-alpine plant species, and more than 70% of them produce primary dormant seeds, exhibiting mainly physiological dormancy (PD) (*Baskin & Baskin, 2014*; *Schwienbacher et al., 2011*). Its production can prevent seed germination during cold autumn/winter seasons and also be seen as an environment that is not long enough to support seedling growth (*Ndihokubwayo, Thang & Cheng, 2016*). Seeds with PD have fully developed embryos that have a physiological inhibiting mechanism (high abscisic acid (ABA)/gibberellic acid (GA) ratio) in the freshly shed seeds (*Baskin & Baskin, 2014*), which prevents germination at any temperature or allows it only at high temperatures (conditional dormancy). ABA induces dormancy during ripening, and GA plays an important role in the liberation of dormancy and the promotion of germination (*Kucera, Cohn & Leubner-Metzger, 2005*), so it demonstrated that artificial addition of gibberellic acid-3 (GA$_3$) could break PD in many plants (*Baskin & Baskin, 2014*). Further, PD is classified into two subclasses and seven levels, non-deep PD (Level 1) is by far the most common, which also includes two sublevels and six types (*Baskin & Baskin, 2021*). Non-deep PD seeds can be released dormant when they are dry-stored at room temperature, a phenomenon researchers called after-ripening (DAR) (*Baskin & Baskin, 2020*). Cold-wet stratification (CS) is also regarded to be an important way to release non-deep PD, especially in seeds of most alpine plants (*Fernandez-Pascual, Jimenez-Alfaro & Bueno, 2017*; *Peng et al., 2021*; *Shimono & Kudo, 2003*; *Yang et al., 2020*), because this is similar to winter cryogenic stimulation seeds that can induce dormant broken seeds to germinate in spring (*Margreiter et al., 2020*). To some extent, as a mediate, CS can promote GA biosynthesis *via* enhanced expression of some specific transcription factors (*Penfield et al., 2005*). As dormancy-break progresses during cold stratification, the ABA/GA ratio decreases (*Finch-Savage & Leubner-Metzger, 2006*),

the range of temperatures at which seeds will germinate, as well as germination rate and percentages, increase (*Baskin & Baskin, 2004*; *Porceddu et al., 2013*).

In addition to being dormant and having a temperature requirement, the responses of seeds to light can also control the timing of germination in alpine regions (*Zhang et al., 2014*). Moreover, the interplay between light requirement and non-deep PD also plays a critical role in regulating the timing of germination (*Nur et al., 2014*; *Jaganathan, Dalrymple & Liu, 2015*; *Peng et al., 2021*). Light requirement prevents germination in deep snow cover or soil and forms a soil seed bank (*Milberg, Andersson & Thompson, 2007*; *Wang et al., 2017*), whereas, the light requirement would be substantially reduced during the low winter temperature in the field or simulating the overwintering environment (CS) in the laboratory, accompanied by dormancy-release (*Peng et al., 2021*; *Wang et al., 2017*). Non-dormant seeds without a light requirement would germinate simultaneously soon after snow melting in spring. Therefore, the timing of germination is regulated through dormancy mechanisms and a light environment, thereby protecting seedling survival and growth.

*Primula* spp., a famous alpine flower belonging to the Primulaceae family, consists of approximately 430 species in the world (*Richards, 1993*). China is the main origin center of *Primula* with approximately 309 species; most are found in high-elevation regions in the Hengduan Mountains (including eastern Tibet, north-western Yunnan, and south-western Sichuan) (*Yang, 2005*). Germination of *Primula* does not generally involve complex dormancy (*Baskin & Baskin, 2004*). Freshly shed seeds of many alpine *Primula* species possess non-deep PD and require special temperature and light combination for germination; these dormancy and germination characteristics delay germination until the following spring and represent an advantageous ecological adaptation toward the unpredictable alpine environments (*Hitchmough, Innes & Mitschunas, 2011*; *Peng et al., 2019a*; *Yang et al., 2020*). *Primula florindae* is an early-flowering perennial forb in subalpine and alpine regions endemic to eastern Tibet, SW China, growing at 2,600 to 4,200 m a.s.l. on streamside, bog margins, wet areas in glades of *Picea* forests or *Rhododendron* thickets. It is a wild ornamental plant, having large, yellow umbels. The high development value of *P. florindae* highlights the increasing need for an investigation of seed germination. Previous studies have focused on reproductive strategies from the flowering to the fruiting stage (*Zhang et al., 2017*; *Zhang et al., 2023*; *Zhang, 2017*); however, few studies have investigated the dormancy and germination of this species.

In this study, we aimed to investigate seed dormancy and germination characteristics of *P. florindae*. Based on the timing of seed germination of *Primula* species in the alpine regions (*Peng et al., 2019a*; *Wang et al., 2017*; *Yang et al., 2020*), we hypothesized that the dormancy mechanism and/or environmental factors (*e.g.*, low temperature and dark) prevent freshly shed seeds of *P. florindae* from germinating after dispersal in autumn, and allow seeds germinate only at the beginning of the growing season in spring. We predicted that winter low temperatures might break dormancy and reduce the high temperature/light requirement for germination, resulting in seed germinating earlier in spring. To test these hypotheses, we asked the following questions: (1) Are freshly shed seeds dormant, and if so, how do dormancy break treatments (GA$_3$, DAR, and CS) release dormancy? (2) How

**Table 1  Population data and seed lot details of *Primula florindae*.**

| Locality | Population code | Location | Altitude (m a.s.l) | Habitat | Vegetation zone | 100 seed weights (g) |
|---|---|---|---|---|---|---|
| Lulang town | LL | 99°74′26.41″E; 29°81′02.65″N | 3,146 | The edge of *Picea* forests | Subalpine | 0.0569 ± 0.002 |
| Sejila Mountain | SJL | 94°69′70.24″E; 29°61′64.12″N | 4,018 | *Rhododendron* thickets | Alpine | 0.0685 ± 0.001 |

do temperature and light affect germination in different states of dormancy? Based on the seed dormancy and germination responses to GA$_3$, temperature, and light measured in the laboratory, we would provide some valuable data for the cultivation of *P. florindae*.

## MATERIALS & METHODS

### Seed material and collection

Freshly-mature fruits of *P. florindae* were collected from two natural populations (Table 1) in early October 2021 at Linzhi region in southeastern Tibet, China. At least 30 capsules were collected from different healthy individual plants (about 10 individuals) by random sampling. After collection, the seeds were removed from non-seed structures, manually cleaned in the laboratory and after 1 week the germination experiments were started. Seeds had been air-dried for 3 months before 100-seed weight was determined.

### Seed germination

For each experiment, three or four replications of 10-20 seeds were placed on 1% water agar in 9-cm-diameter plastic Petri dishes, which were put into transparent plastic bags to prevent desiccation. The plastic bags with the dishes of seeds in them were placed on shelves in incubators at the corresponding temperature, where their positions change daily after inspection. The criterion for germination was visible radicle protrusion (*Peng et al., 2021*). Seeds incubated in light were monitored daily and germinated seeds were discarded, while dark-incubated seeds were counted only at the end of the test to avoid any exposure to light (*Peng et al., 2023*). The germination test lasts for at least 28 days with the viability of the ungerminated seeds checked at the end of the germination test. Seeds with fresh and healthy tissues (a plump, firm, and white embryo and endosperm) were considered viable, and with moldy, necrotic, and soft tissues were nonviable (*Sacco et al., 2021*).

### Germination of fresh seeds

Seeds were incubated at seven constant temperatures (CT; 1, 5, 10, 15, 20, 25, and 30 °C) and three alternating temperature regimes (AT; 5/1, 15/5, and 25/15 °C) in a photoperiod of 12/12 h (light/dark, hereafter light). Photon irradiance (400–700 nm) from cool white fluorescent tubes during the light period across all temperatures was 2500 lux (*Peng et al., 2023*). To verify germination in constant darkness, Petri dishes with seeds need to be parceled with two layers of aluminum foil before incubating at 1, 5, 15, 25, 5/1, 15/5, and 25/15 °C. The constant (1−20 °C) and alternating temperatures (5/1, 15/5, and 25/15 °C) represent the daily temperature variation in the soil surface layer during the growing

season (late April to late August) in the Sejila Mountains (*Liu, Liang & Zhu, 2011*; *Wang et al., 2017*). 25−30 °C represents the extremely high temperature range during the growing season.

### Dormancy breaking experiments

The effects of gibberellic acid (GA$_3$) on germination of fresh seeds were tested. Seeds were incubated in Petri dishes with 1% water agar containing two concentrations of GA$_3$ (0 (control), 20 or 200 mg L$^{-1}$) at 15/5 and 25/15 °C.

To determine the effect of dry after-ripening (DAR) on dormancy-break and germination, seeds were stored dry in a paper bag at ambient room condition (25–55% relative humidity; 13−22 °C). Once the seeds have been treated with DAR treatments (D3 and D6, 3 and 6 months respectively) they are incubated under the same temperature and light conditions as the fresh seeds described above.

To demonstrate the effect of cold-wet stratification (CS), the seeds were placed evenly on 1% water agar in Petri dishes, which were wrapped with two layers of aluminum foil and placed in a refrigerator at 1 °C to simulate the natural temperature under snow in winter (*Wang et al., 2017*). After 3 and 6 months of CS treatments (C3 and C6, respectively), seeds are incubated in the same way as the fresh seeds mentioned above.

### Germination characteristics analysis

The final germination percentage (GP) and the mean germination time (MGT) were calculated as follows: GP $= \sum$Gi/N; MGT (days) $= \sum$(i × Gi)/ $\sum$Gi, where i signifies the day of germination, counted since the day of sowing, Gi represents the number of seeds germinated on the ith day and N denotes the total number of filled seeds (*Wei et al., 2020*).

### Statistical analysis

Statistical analysis was performed utilizing the Statistical Package for Social Science version 20.0 (SPSS, Chicago, IL, USA). Generalized linear models (GLMs) were used to compare: (1). whether the different GA$_3$ concentrations (20 and 200 mg L$^{-1}$) treatments could significantly increase the GP of fresh seeds in light and dark under two alternating temperatures (15/5 and 25/15 °C); (2). whether DAR and CS treatments significantly increased GP and/or decreased MGT, relative to fresh seeds in different light and temperature treatments (*Peng et al., 2021*). The *post-hoc* pairwise comparison *t*-test (using Bonferroni adjustment) was employed to compare the mean values of the treatment-related response variables (*Peng et al., 2023*). All data were expressed as mean $\pm$ SE. All figures were drawn with Origin 2021.

## RESULTS

### Effect of GA$_3$ and light condition on germination of fresh seeds

For two populations, germination of fresh seeds in water in light was significantly higher than it was in dark at 15/5 and 25/15 °C (Fig. 1). GA$_3$ concentration of 20 mg L$^{-1}$ significantly ($p < 0.01$) improved germination, except the SJL in light and the LL in dark at 15/5 °C. GA$_3$ concentration of 200 mg L$^{-1}$ had the highest germination (>80%) at both

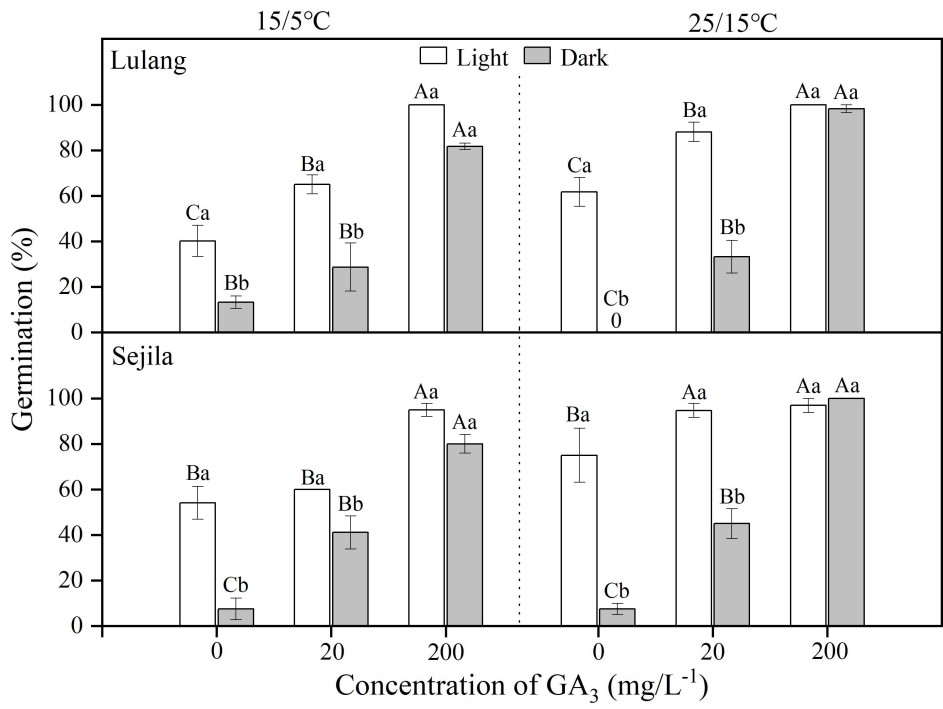

**Figure 1** **Effect of GA$_3$ and light condition on germination of fresh seeds.** Germination percentages of fresh seeds of *Primula florindae* at two alternating temperatures (15/5 and 25/15 °C) in light and in dark in various concentrations of GA$_3$. 0 means the treatment has no germination. Different uppercase letters indicate significant differences ($p < 0.05$) across all concentrations of GA$_3$ for the same light condition and different lowercase letters significant differences between light conditions at the same concentration.

**Table 2** **GLM analysis of light condition and GA$_3$ concentrations and their interactions on final germination percentages of *Primula florindae* (LL and SJL) at two alternating temperatures.**

| Population code | LL | | | | SJL | | | |
|---|---|---|---|---|---|---|---|---|
| Treatments | 15/5 °C | | 25/15 °C | | 15/5 °C | | 25/15 °C | |
| | *df* | *p* | *df* | *p* | *df* | *p* | *df* | *p* |
| Light condition (*L*) | 1 | <0.000 | 1 | <0.000 | 1 | <0.000 | 1 | <0.000 |
| GA$_3$ concentration (*G*) | 2 | <0.000 | 2 | <0.000 | 2 | <0.000 | 2 | <0.000 |
| *L* × *G* | 2 | 0.426 | 2 | <0.000 | 2 | <0.000 | 2 | <0.000 |

**Notes.**
Extremely significant difference: $P < 0.001$; No significant difference: $P > 0.05$.

temperatures, regardless of light condition; moreover, there was no difference between germination in light and dark (Fig. 1). GLMs showed that germination of fresh seeds was significantly stimulated by GA$_3$ concentration ($p < 0.001$), light ($p < 0.001$) and the interaction between light and GA$_3$ concentration ($p < 0.001$), except light × GA$_3$ concentration at 15/5 °C for LL (Table 2).

### Effects of temperatures and light condition on germination of fresh seeds

Fresh seeds from the two *P. florindae* populations germinated well (LL: >60%; SJL: >70%) at the warmer temperatures (20, 25, and 25/15 °C) in light, but only about 40% at 10 and 15/5 °C or even to 0% germination below 10 °C (Fig. 2). Fresh seeds incubated in dark did not germinate as effectively as in light, and germination was lower than 30% or even 0% at the temperature at which the test was performed (1, 5, 15, 25, 5/1, 15/5 and 25/15 °C; Figs. 3 and 4).

### Effect of DAR treatments and light condition on seed germination at different temperatures

After 3 and 6 months of DAR treatments in both two populations, GP generally was higher than those of fresh seeds in light at all constant (except at 1, 20−30 °C) and alternating temperature regimes (Fig. 3). However, DAR treatments did not promote or promote weakly seed germination in dark ($p < 0.05$, Fig. 3). Dry treatments also significantly decreased MGT in light except at 25 °C for LL at 20, 25, and 25/15 °C for SJL (Fig. 3). The fitted GLMs results showed that DAR treatments, and light condition factors, and their interactions had a statistically significant effect on GP and MGT at CT and AT (Tables 3 and 4) in both two populations (except the dry after-ripening at CT for SJL).

### Effect of CS treatments and light condition on seed germination at different temperatures

Compared to fresh seeds in both two populations, CS seeds had higher GP and/or lower MGT at all constant temperatures (except at 1 °C, no germination occurred at this temperature) and alternating temperature regimes, regardless of light condition (Fig. 4). CS treatments for 3 and 6 months promoted more seed germination in light than in dark, especially at the lower temperatures (5 and 5/1 °C), but widened the temperature range of germination in dark. The fitted GLMs results showed that CS treatments, light condition factors, and their interactions had a statistically significant effect on GP and MGT at CT and AT (Tables 3 and 4) in both populations.

## DISCUSSION

*P. florindae* is a perennial herb endemic to the alpine habitat in eastern Tibet, but information on seed dormancy and germination of this species is scarce. *Hitchmough, Innes & Mitschunas (2011)* reported that surface sowing (a thin layer of soil covering) was sufficient to stimulate germination of *P. florindae* in dry-stored seed, but they did not provide any information about seed dormancy and germination. In this study, we determined how GA₃, light, temperature, CS, and DAR treatments affect seed germination by conducting a series of laboratory experiments. We found that the autumn-dispersal seeds did not germinate or germinate rarely at low temperatures, especially in dark condition (Figs. 2–4). After overwintering, dormancy release and non-dormant seeds germinate at a wide range of temperatures, regardless of light condition (Fig. 4). Our results showed that freshly-matured seeds with PD and light requirements for germination, synchronized the

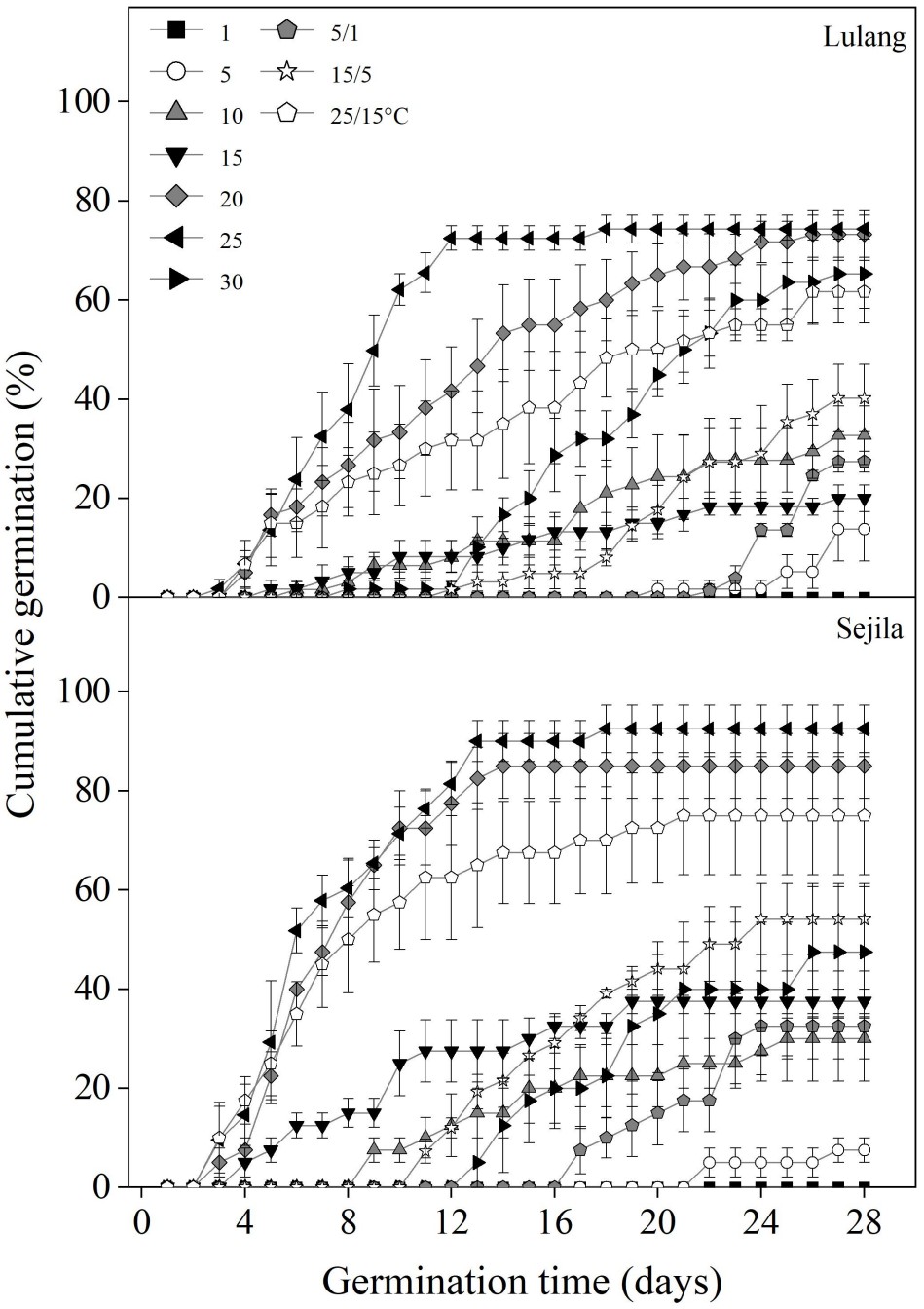

**Figure 2  Effect of temperatures on cumulative germination of fresh seeds.** Cumulative germination percentages of fresh seeds in Lulang and Sejila seed populations at seven constant and three alternating temperatures under light conditions. Scatters represent standard error of the mean.

timing of germination of *P. florindae* in the alpine habitats to early spring. This finding confirms that *P. florindae* seeds have PD and supports earlier observations of other alpine species (*Peng et al., 2017*; *Peng et al., 2018*; *Peng et al., 2019a*; *Peng et al., 2019b*).

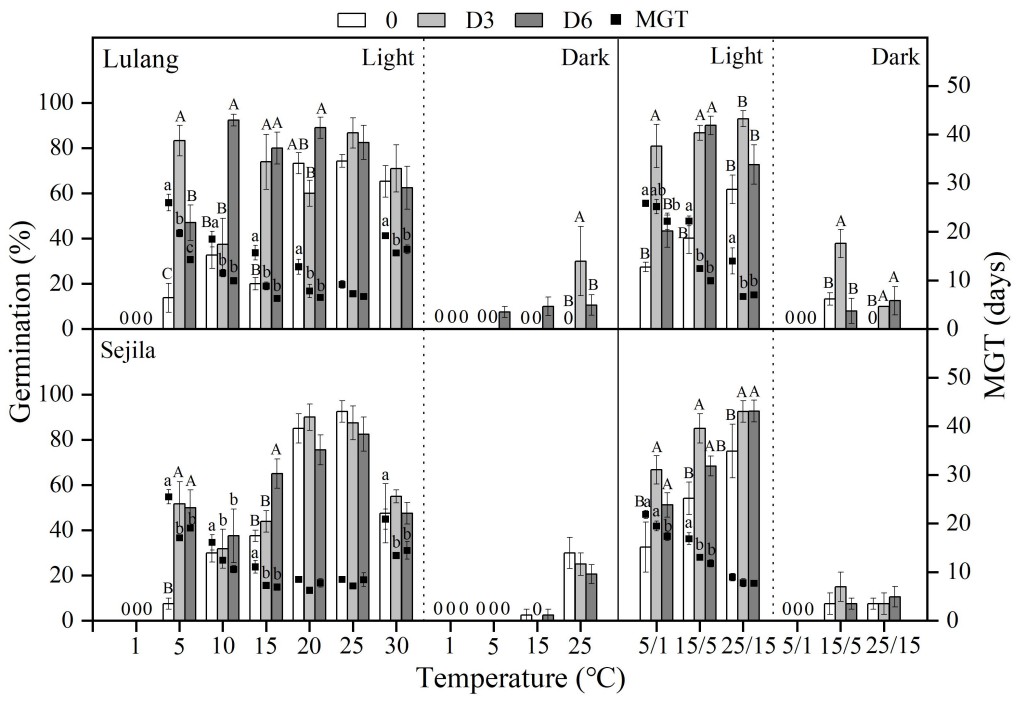

**Figure 3  Effect of DAR treatments and light condition on seed germination at constant and alternating temperatures.** Final germination percentage (GP) and mean germination time (MGT) at each constant and alternating temperature with respect to dry after-ripened (DAR) period (Fresh 0, control; D3 and D6, DAR at room temperature for 3 and 6 months, respectively) in Lulang and Sejila seed populations under light and dark conditions. 0, no seeds germinated in this treatment. Bars and scatters represent standard error of the mean. Bars with different uppercase letters are differ significantly ($p < 0.05$) for GP at the same temperatures of the three DAR periods. Scatters with different lowercase letters are differ significantly ($p < 0.05$) for MGT at the same temperatures of the three DAR periods under light condition. The unmarked bars and scatters showed no significant difference.

In regular PD, there are three levels (non-deep, intermediate, and deep), and there are six types of non-deep PD (*Baskin & Baskin, 2021*). The dormancy type of *P. florindae* has been demonstrated in three steps. First, freshly-matured seeds had fully developed embryos, and germination was significantly improved by GA₃ at 15/5 and 25/15 °C (Fig. 1). These results suggested that shed seeds may exhibit PD and supported earlier observations of other *Primula* species (*Baskin & Baskin, 2014*; *Hitchmough, Innes & Mitschunas, 2011*). Second, a period of DAR and CS treatments could break dormancy (Figs. 3 and 4). Thus, these results indicated that seeds appeared to have non-deep PD. Further, fresh seeds could germinate well at higher temperatures but not at lower temperatures. As dormancy break treatments (DAR and CS) in the seed populations continue, seeds gained the ability to germinate over a wide range of temperatures (Figs. 3 and 4), especially decreasing in the minimum temperatures in dark at which seeds could germinate (Figs. 3 and 4), indicating that freshly shed seeds of *P. florindae* have type 2 non-deep PD (*Baskin & Baskin, 2014*). Seeds with type 2 non-deep PD first enter conditional dormancy and germinate to high

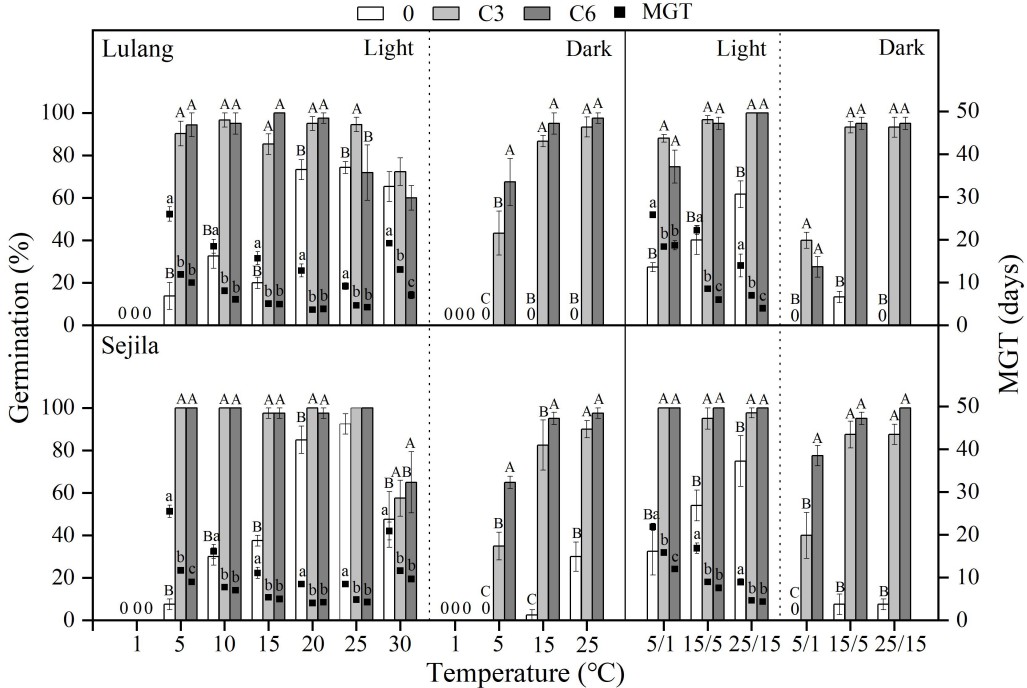

**Figure 4** **Effect of CS treatments and light condition on seed germination at constant and alternating temperatures.** Final germination percentage (GP) and mean germination time (MGT) at each constant and alternating temperature with respect to cold stratification (CS) period (Fresh 0, control; C3 and C6, CS at room temperature for 3 and 6 months, respectively) in Lulang and Sejila seed populations under light and dark conditions. 0, no seeds germinated in this treatment. Bars and scatters represent standard error of the mean. Bars with different uppercase letters are differ significantly ($p < 0.05$) for GP at the same temperatures of the three CS periods. Scatters with different lowercase letters are differ significantly ($p < 0.05$) for MGT at the same temperatures of the three CS periods under light condition. The unmarked bars and scatters showed no significant difference.

percentages only at high temperatures, and the lowest temperature at which they will germinate decreases as dormancy break progresses (*Soltani, Baskin & Baskin, 2017*).

Matured seeds of *P. florindae* with type 2 non-deep PD had high temperatures temperature requirements for germination (Figs. 3 and 4). So, freshly shed seeds likely failed to germinate immediately after dispersal in autumn due to low soil temperatures. However, the timing of germination in this species was regulated by seasonal temperature changes. Non-deep PD in fresh seeds was gradually broken in CS treatments (3 and 6 months) at winter temperatures under snow (1 °C, Fig. 4), resulting that the lowest temperature at which they will germinate decreases in the field during winter (Fig. 4). Therefore, dormancy break at low temperature is completed by late winter, and a high percentage of them germinate in a short period over the widest range of temperatures (Fig. 4) in spring. Non-dormant seeds can germinate soon after snowmelt at the beginning of the short growing season, which maximizes the length of the seedling's growing season.

In addition to dormancy break at low temperatures, germination of *P. florindae* seeds also depends on the light condition (Figs. 3 and 4). *Hitchmough, Innes & Mitschunas*

**Table 3** GLM analysis of light condition, dormancy breaking treatments (dry after-ripening and cold-wet stratification) and their interaction on germination percentages of *Primula florindae* (LL and SJL) at constant (CT) and alternating (AT) temperatures.

| Population code | LL | | | | SJL | | | |
|---|---|---|---|---|---|---|---|---|
| Treatments | CT | | AT | | CT | | AT | |
| | *df* | *p* | *df* | *p* | *df* | *p* | *df* | *p* |
| Light condition (*L*) | 1 | <0.000 | 1 | <0.000 | 1 | <0.000 | 1 | <0.000 |
| Dry after-ripening (*D*) | 2 | <0.000 | 2 | <0.000 | 2 | 0.127 | 2 | <0.000 |
| Cold-wet stratification (*C*) | 2 | <0.000 | 2 | <0.000 | 2 | <0.000 | 2 | <0.000 |
| *L* × *D* | 2 | <0.000 | 2 | <0.000 | 2 | 0.011 | 2 | <0.000 |
| *L* × *C* | 2 | <0.000 | 2 | <0.000 | 2 | <0.000 | 2 | <0.000 |

Notes.
Extremely significant difference: $P < 0.001$; Significant difference: $P < 0.05$; No significant difference: $P > 0.05$.

**Table 4** GLM analysis of dry after-ripening and cold-wet stratification on mean germination time (MGT) of *Primula florindae* (LL and SJL) at constant (CT) and alternating (AT) temperatures in light.

| Population code | LL | | | | SJL | | | |
|---|---|---|---|---|---|---|---|---|
| Treatments | CT | | AT | | CT | | AT | |
| | *df* | *p* | *df* | *p* | *df* | *p* | *df* | *p* |
| Dry after-ripening (*D*) | 2 | <0.000 | 2 | <0.000 | 2 | <0.000 | 2 | <0.000 |
| Cold-wet stratification (*C*) | 2 | <0.000 | 2 | <0.000 | 2 | <0.000 | 2 | <0.000 |

Notes.
Extremely significant difference: $P < 0.001$.

*(2011)* reported that dry-storage seeds of *P. florindae* could not be germinated well when sown in the soil at 10 mm depth, indicating that the light requirement prevents seed germination in deep soil. However, we do not know how light requirement change with dormancy broken in *P. florindae*. Our study showed that fresh seeds incubated in dark did not germinate as well as those in light (Figs. 3 and 4). Dry-storage (DAR) for 3 and 6 months broken seed dormancy and increased seed germination percentages in light, but DAR did not significantly improve germination in dark except at warmer temperatures (25 and 25/15 °C) for LL (Fig. 3 and Table 3). The light requirement for germination varies with the season (*Pons, 2000*) in many alpine plants with non-deep PD, and it can be reduced by CS treatments (*Peng et al., 2021*; *Shimono & Kudo, 2005*). By comparing germination between the daily photoperiod and continuous darkness, we found that the light requirement changed as seeds of *P. florindae* came out of dormancy. CS treatment for 3 and 6 months (Fig. 4), treatment with >20 mg L$^{-1}$ GA$_3$ (Fig. 1) changed the light requirement for germination of *P. florindae* seeds. Thus, the seeds can germinate in dark but only in spring during a short period of the year (*Wang et al., 2017*). Moreover, most of the non-dormant seeds (CS seeds) of both populations could germinate in dark, thus *P. florindae* should not be able to form a large persistent soil seed bank. The similar light sensitivity in the germination of seeds of the two populations may be a result of adaptive selection in species exposed to similar micro-habitat in the alpine regions.

Based on our results, we suggest the following two scenarios for the timing of *P. florindae* germination in nature. First, at the time of dispersal in autumn (late September-October),

seeds of *P. florindae* can germinate well at relatively warmer (>20 °C) temperatures (Fig. 2). However, temperatures in its high elevation habitat are too low (see Figure 5a in *Wang et al. (2017)*) for germination. During winter, cold stratification reduces the temperature requirement for germination. Thus, the germination temperature of cold-stratified seeds widens from high to low temperature (Fig. 4). Such temperature requirements for germination would prevent *P. florindae* seeds germinating in autumn because the high temperature requirement for germination is not being met, and allows them rapidly germination soon after snowmelt in spring, which undoubtedly increases the chance for survival of seedlings during their first establishment year. Second, *P. florindae* have relatively small seeds (Table 1); there are approximately 1,500 seeds per gram. So, fresh seeds were easily covered by soil or a deep layer of litter and snow. Moreover, fresh seeds had strict light requirement, especially at lower (1−5 °C) temperatures; thus, any seeds in the absence of light could not germinate in autumn (Fig. 3). CS treatments at winter temperatures reduced the light requirement for germination (Fig. 4) and promoted germination of a high number of buried seeds in spring. The ability of *P. florindae* seeds to germinate over the wide range of temperatures without a strict light germination in early spring ensures a sufficient length of the growing season for seedling recruitment and the plants to grow to a large enough size to survive over winter.

## CONCLUSIONS

Matured seeds of *P. florindae* have type 2 non-deep PD and light requirement for germination, which prevent seeds from germinating after dispersal in autumn. Seeds lost dormancy and light requirement for germination in winter temperatures and many of them germinated at low temperatures in spring when temperature, light, and moisture conditions are favorable for seedling establishment and growth, enabling plants to take full advantage of the short, unpredictable alpine growing season. Thus, our hypothesis that seed dormancy and special germination requirement restrict germination to the beginning of the growing season is supported. Moreover, from farm cultivation, it is suggested that seeds germinate following DAR and CS treatments at higher temperatures (20−25 °C) in light conditions.

## ACKNOWLEDGEMENTS

The authors are grateful to Jing You, Wanning Zheng, Aiyu Yang, Min Weng, Li Yang, Yuan Liu, Lingfang Xu, and Yangqian Zhang for their assistance in laboratory work.

### Funding

This study was supported by the Second Tibetan Plateau Scientific Expedition and Research (STEP) program (2019QZKK0502 to Hang Sun), and the National Natural Science Foundation of China (grants 31700284 and 32060079 to Deli Peng). The funders had no

role in study design, data collection and analysis, decision to publish, or preparation of the manuscript.

## Grant Disclosures
The following grant information was disclosed by the authors:
Second Tibetan Plateau Scientific Expedition and Research (STEP) program: 2019QZKK0502.
National Natural Science Foundation of China: 31700284, 32060079.

## Competing Interests
The authors declare there are no competing interests.

## Author Contributions
- Yingbo Qin conceived and designed the experiments, performed the experiments, analyzed the data, prepared figures and/or tables, authored or reviewed drafts of the article, and approved the final draft.
- Boyang Geng performed the experiments, prepared figures and/or tables, and approved the final draft.
- Li-E Yang performed the experiments, authored or reviewed drafts of the article, and approved the final draft.
- Deli Peng conceived and designed the experiments, performed the experiments, authored or reviewed drafts of the article, and approved the final draft.

## Data Availability
The raw data are available in the Supplemental Files.

## Supplemental Information
Supplemental information for this article can be found online at http://dx.doi.org/10.7717/peerj.15234#supplemental-information.

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
