# Peer review of "Non-deep physiological dormancy and germination characteristics of Primula florindae (Primulaceae), a rare alpine plant in the Hengduan Mountains of southwest China"

_PeerJ, doi:10.7717/peerj.15234_

## Round 0.1 · original submission · Major Revisions

Dear author, considering the report of the reviewers, your article can not be published in its current form.

However, the Section Editor noted that

> I disagree with the reviewer's statement that 400 seeds are required for a germination test. That may be the standard for crop plant agricultural tests but the literature relevant to this type of study (see for example papers by Kathleen Donohue, Leonie Bentsink, or Johanna Schmitt) have a study size similar to this


Reviewer 1 ·

Basic reporting

Not clear about the topic
confusing statements given
Unstructured methodologies followed

Experimental design

Research and questions are not well defended
Meaningful methodologies and statistical methods are not adopted

Validity of the findings

.

Additional comments

Abstract
It should give message about carried experiment in proper channel and flow of work which is missing in writing. One or few sentences are needed for methodologies. Some important experimental findings should be included in the abstract with proper reasoning.
Introduction
The subject of manuscript is remarkably interesting, but author should more relay on the subject of manuscript. They give classification of all kinds of dormancy which is not needed. They must be described only non-deep physiological dormancy. Why it is important? What is the actual mechanism? Where it is found. Like that. Introduction part explained only about dormancy but not for the Primula florindae, its importance in forest, its overall distribution etc. This is an insufficient information for dormancy as well as for Primula florindae. Objective of the study are unclear. Overall introduction has very confusing statements without proper formation.

Line no 43: change the first sentence. Start with positive sentence
Line no 53: remove “environmental” word
Line no 55: Change the Defination of dormancy
Line no 61: Baskin and Baskin give classification in 1998 and 2004.
Line no 67 to 69: confusing statement
Line no 78: Reframe the sentence
Line no 86: add “family” after “flowers belonging to the….”
Line no 88: Grammatically incorrect sentence
Line no 105 to 107: Reframe the objective of the study
What is non deep physiological dormancy?
What is the history and how it works in alpine regions?
Who classify this (non-deep physiological dormancy) in which category?
What is the relation of light and non-deep physiological dormancy?

Materials and methods
Materials and methods written unambiguously. Aim of study cannot be addressed without proper methodology and description of materials. Pictorial proof is needed as a supplementary material.
Line no 125: Unclear
Line no 142: Incomplete heading
Line no 158: Give citation
Line no 162: Give citation who suggested it?
How authors select constant temperatures? Cite the methodology or describe the temperature and conditions and assumptions of selecting specific temperature. Similarly, describe about all experiments regarding temperature.
Results
Authors working with two different collections but how these collections are different from each other is not mentioned. It is recommended to add in the study. Graphical results are easy to understand but to prove its actual effect one or more comparative photographs of experiment is mandatory to increase effectiveness of carried experiment.
Line no 176: correct the unit as 200 mg L-1
Line no 180: Remove ( ) from “except light × GA3 concentration at 15/5°C for LL)”
Discussion
It is written well but it needs to improve by giving perfect reasons of results with proper citations. Given citations must be explained with their results and conclusions. Pictorial presentation may increase the understanding of experiments.
Conclusions
Conclusion only contain results of the experiment. It should have novelty in findings. It may include some recommendations regarding preservation of the species. Novelty and proper findings are missing from the conclusion.

Annotated reviews are not available for download in order to protect the identity of reviewers who chose to remain anonymous.

Reviewer 2 ·

Basic reporting

The authors investigated seed dormancy and germination characteristics of an alpine plant Primula florindae in Southwest China. Seed dormancy is an important mechanism for ensuring seedling establishment in the right place and time. The results from this study could be of interest to those working on seed ecology in the alpine ecosystem.

Experimental design

See additional comments.

Validity of the findings

No comment.

Additional comments

Introduction. In the last paragraph, the first question includes two parts that should be combined.
Materials & Methods. More detailed information is needed. For example, how many individuals are the capsules collected from, and can they represent the tow populations? How long were the seeds air-dried? It is important because dry storage can affect seed dormancy. Are the constant and alternating temperatures representative the seasonal temperatures in the natural habitat?
In addition, the reference for calculating GP and MGT should be provided.
No information for the vegetation of the two populations (LL and SJL) is provided (although Table 1 shows the location and altitude).
Results. The description in line 185-187 cannot been seen in Fig. 3-6.
Discussion. Line 212, unclear about ‘surface sowing’.
Lines 214-217 is not suitable for the discussion.
Line 235, need to provide more detailed information for type 2 non-deep PD.
In all figures, need to give the full names for LL and SJL in the legend.
Fig. 3 and 4 (also Fig. 5 and 6) should be combined, because the two figures show the same information.
English needs to be improved. I noted some errors below (only for the Abstract).
Line 21, ‘conditions’ should be ‘probability’.
Line 24, ‘Xizang’ should be ‘Tibet’.
Line 26, unclear about ‘the first opportunity’.
Line 28, ‘we-re’ should be ‘were’.
Line 33, ‘It’ should be ‘Our results’. ‘undergo’ should be ‘have’.
Line 34, delete ‘in’.

Reviewer 3 ·

Basic reporting

There are some unclear statements was raise in introduction part as well in materials and methods section.

Experimental design

No. of replication used in experiment should be very specific.

Validity of the findings

No comment

Additional comments

There is a one major mistake was done in this experiment that was number of seed used in the experiment. As per the ISTA, minimum seed required for germination test is 400.

Annotated reviews are not available for download in order to protect the identity of reviewers who chose to remain anonymous.

---

## Round 0.2 · Minor Revisions

Dear authors

As noted by Reviewer 1 you have not provided a formal response to their comments. Please resubmit your article, this time including a full response letter dealing with the comments of the first reviewer on the original submission.

Reviewer 2 ·

Basic reporting

I cannot find a point-to-point response to my previous comments.

Experimental design

See above.

Validity of the findings

See above.

Reviewer 3 ·

Basic reporting

The experiment is very basic and finding is cleared. Authors has been incorporated and modified as per suggested.

Experimental design

No comment

Validity of the findings

The finding results are well explained. Tables and figures well stated.

Additional comments

The experimental finding could be use to propagate such of type of alpine forest plant by breaking the seed dormancy.
It's very much necessary to understand the type of dormancy and every seed require a very distinct set of condition for germination.
So, the present result will helps in directly or indirectly in conservation of plant genetic resources.

---

## Round 0.3 · accepted · Accept

Dear authors, congratulations, your paper has been accepted.

Reviewer 2 ·

Basic reporting

The authors have revised the manuscript according to my previous comments. I only have some minor comments for language.
Line 24, delete ‘in alpine plants’.
Line 133, delete ‘entirely’.
Lines 209-211, confusing sentence.
Line 219, delete the first ‘and’.
Line 246, delete ‘regular’.
Line 322, clarify ‘farm cultivation’.

Experimental design

No additional comments.

Validity of the findings

No additional comments.